Current trends in the epidemiology of multidrug-resistant and beta-lactamase-producing Pseudomonas aeruginosa in Asia and Africa: a systematic review and meta-analysis

http://orcid.org/0000-0003-4069-9381 Salleh Mohd Zulkifli m.z.salleh@usm.my
http://orcid.org/0000-0002-0009-2772 Nik Zuraina Nik Mohd Noor
http://orcid.org/0000-0003-3344-1518 Deris Zakuan Zainy
Mohamed Zeehaida zeehaida@usm.my
Department of Medical Microbiology & Parasitology, School of Medical Sciences, Universiti Sains Malaysia , Kota Bharu, Kelantan , Malaysia
Tharmalingam Nagendran
Electronic publication date: 2025 Feb 24
Publication date: 2025
Volume: 13
Electronic Location ID: e18986
Received 2024 Sep 4; Accepted 2025 Jan 22
Copyright: © 2025 Salleh et al.
Copyright year: 2025
Copyright holder: Salleh et al.
License: This is an open access article distributed under the terms of the Creative Commons Attribution License, which permits unrestricted use, distribution, reproduction and adaptation in any medium and for any purpose provided that it is properly attributed. For attribution, the original author(s), title, publication source (PeerJ) and either DOI or URL of the article must be cited.
License URL: https://creativecommons.org/licenses/by/4.0/

Keywords: Pseudomonas aeruginosa, Multidrug resistance, Extended-spectrum beta-lactamase, Metallo-beta-lactamase, Epidemiology, Systematic review, Meta-analysis

Funding: Universiti Sains Malaysia Research University 1001/PPSP/8012351 This work was supported by the Universiti Sains Malaysia Research University Grant 1001/PPSP/8012351. The funders had no role in study design, data collection and analysis, decision to publish, or preparation of the manuscript.

==============================
Pseudomonas aeruginosa continues to be a significant contributor to high morbidity and mortality rates worldwide, particularly due to its role in severe infections such as hospital-acquired conditions, including ventilator-associated pneumonia and various sepsis syndromes. The global increase in antimicrobial-resistant (AMR) P. aeruginosa strains has made these infections more difficult to treat, by limiting the effective drug options available. This systematic review and meta-analysis aim to provide an updated summary of the prevalence of AMR P. aeruginosa over the past 5 years. A systematic search was performed across three major electronic databases—PubMed, ScienceDirect, and Web of Science—yielding 40 eligible studies published between 2018 and 2023. Using a random-effects model, our meta-analysis estimated that the overall prevalence of P. aeruginosa in Asia and Africa over the past 5 years was 22.9% (95% CI [14.4–31.4]). The prevalence rates for multidrug-resistant (MDR) and extensively drug-resistant (XDR) P. aeruginosa strains were found to be 46.0% (95% CI [37.1–55.0]) and 19.6% (95% CI [4.3–34.9]), respectively. Furthermore, the prevalence rates of extended-spectrum β-lactamase- and metallo-β-lactamase-producing P. aeruginosa were 33.4% (95% CI [23.6–43.2]) and 16.0% (95% CI [9.8–22.3]), respectively. Notably, resistance rates to β-lactams used for treating pseudomonal infections were alarmingly high, with rates between 84.4% and 100.0% for cephalosporins, and over 40% of P. aeruginosa isolates showed resistance to penicillins. Our analysis identified the lowest resistance rates for last-resort antimicrobials, with 0.3% (95% CI [0.0–1.3]) resistance to polymyxin B and 5.8% (95% CI [1.5–10.2]) to colistin/polymyxin E. The low resistance rates to polymyxins suggest that these antibiotics remain effective against MDR P. aeruginosa. However, the findings also highlight the critical public health threat posed by antimicrobial-resistant P. aeruginosa, particularly concerning β-lactam antibiotics. This underscores the need for effective and carefully planned intervention strategies, including the development of new antibiotics to address the growing challenge of resistance. Developing robust antibiotic treatment protocols is essential for better management and control of pseudomonal infections globally. Therefore, continued research and international collaboration is vital to tackle this escalating public health challenge. This study protocol was registered with the International Prospective Register of Systematic Reviews (PROSPERO), under registration number CRD42023412839.

Introduction

Pseudomonas aeruginosa is a Gram-negative bacterium that frequently causes infections, particularly in immunocompromised individuals and patients with cystic fibrosis (CF). It is a member of the ESKAPE group, which includes six highly virulent bacteria (Enterococcus faecium, Staphylococcus aureus, Klebsiella pneumoniae, Acinetobacter baumannii, P. aeruginosa, and Enterobacter spp.) (De Oliveira David et al., 2020), notorious for their increasing resistance to antibiotics. These pathogens are not only becoming more prevalent but are also evolving new resistance mechanisms, which significantly complicate treatment efforts.

Antibiotic-resistant P. aeruginosa is commonly linked to healthcare-associated infections (HAIs), especially in intensive care units (ICUs), surgical wards, and long-term care facilities. A point prevalence survey conducted across 28 European countries during 2016–2017 found that P. aeruginosa was the fifth most frequent cause of hospital-acquired infections (HAI), with a prevalence rate of 7.1% in tertiary care hospitals (Suetens et al., 2018). Multidrug-resistant (MDR) and extensively drug-resistant (XDR) P. aeruginosa present serious challenges in healthcare settings due to their broad resistance to multiple antibiotic classes, leading to higher rates of morbidity, mortality, and increased healthcare costs. In certain regions, the prevalence of MDR and XDR P. aeruginosa ranges between 15% and 30%, and in ICUs, it can reach up to 48.7%, due to factors such as the increased use of broad-spectrum antibiotics such as aminoglycosides, carbapenems, and cephalosporins, increased patient vulnerability, and prolonged hospital stays (Horcajada et al., 2019; Ribeiro et al., 2019). The increasing prevalence of MDR and XDR P. aeruginosa in various parts of the world, including Europe, China, and some Southeast Asian countries, can be attributed to factors such as the widespread misuse of antibiotics, insufficient infection control measures, and the pathogen’s capacity to employ various resistance strategies, which include the presence of mobile genetic elements that harbor resistance genes (Karruli et al., 2023).

MDR and XDR P. aeruginosa display a variety of resistance mechanisms, involving chromosomal factors and complex regulatory pathways that control intrinsic, acquired, and adaptive resistance. Key mechanisms include the production of β-lactamases like carbapenemases, extended-spectrum β-lactamases (ESBLs), metallo-β-lactamases (MBLs), and the expression of AmpC cephalosporinase (Glen & Lamont, 2021). The inducible production of β-lactamases, particularly in response to aminopenicillins and certain cephalosporins, contributes to reduced susceptibility to these antibiotics, including imipenem. P. aeruginosa also possesses active efflux pump systems, such as the MexAB-OprM and inducible MexXY efflux pumps, which lower susceptibility to a wide range of antibiotics, including β-lactams, fluoroquinolones, and chloramphenicol, while contributing to inherent resistance to aminoglycosides (Avakh et al., 2023). Moreover, whole-genome screenings have revealed the presence of mobile genetic elements and numerous resistance genes, adding to P. aeruginosa’s mutational resistome (López-Causapé et al., 2018). In addition, P. aeruginosa has the ability to thrive in metal-scarce conditions within the host by producing metallophores, which enable prolonged bacterial survival and potentially enhance resistance to antibiotics (Ghssein & Ezzeddine, 2022). The acquired resistance in P. aeruginosa can result from either horizontal gene transfer or mutational changes, while adaptive resistance is associated with biofilm formation in the lungs of infected patients, where the biofilm acts as a barrier, limiting antibiotic penetration to the bacterial cells (Pang et al., 2019).

Implementing robust surveillance programs and antibiotic stewardship initiatives is essential to combat the spread of antibiotic-resistant P. aeruginosa and minimize the impact of MDR and XDR P. aeruginosa infections on patients and the healthcare system (Horcajada et al., 2019; Glen & Lamont, 2021). These efforts are essential for monitoring resistance trends, identifying outbreaks, and promoting the rational use of antibiotics to preserve the efficacy of current antimicrobial agents. This systematic review and meta-analysis (SRMA) seek to provide a comprehensive, up-to-date prevalence estimate of antibiotic-resistant P. aeruginosa based on data published in the last 5 years.

Methodology

Search strategy and selection criteria

This SRMA was conducted based on the PRISMA guidelines (Page et al., 2021). The study protocol was registered with the International Prospective Register of Systematic Reviews (PROSPERO), under registration number CRD42023412839. A thorough literature search was conducted between March 2023 and April 2023 to identify studies on the prevalence of antimicrobial-resistant (AMR) P. aeruginosa, using the PubMed, ScienceDirect, and Web of Science databases (Fig. 1). Relevant search terms and keywords used included “Pseudomonas aeruginosa AND drug resistance AND extended-spectrum beta-lactamase”, “Pseudomonas aeruginosa AND drug resistance AND metallo-beta-lactamase”, and “Pseudomonas aeruginosa AND drug resistance AND ESBL”. Additionally, reference lists of the selected articles were reviewed to find further relevant studies.

Figure 1 A PRISMA flow diagram depicting the study selection process and literature search outcomes.

Three distinct web databases (PubMed, ScienceDirect, and Scopus) were employed to search for eligible studies reporting on antimicrobial-resistant P. aeruginosa through predefined search strategies. In total, 7,343 records were obtained and duplicates were eliminated using the EndNote 20 software. Subsequently, these records underwent screening against predefined inclusion criteria before their inclusion in this systematic review and meta-analysis.

Inclusion and exclusion criteria

To be included in this SRMA, studies needed to provide sufficient data on the prevalence of antimicrobial-resistant P. aeruginosa from all countries, regardless of gender or age. Inclusion criteria required that studies be full-length original research articles published in English between 2018 and 2023, ensuring up-to-date information. Excluded from the analysis were studies that did not report data on P. aeruginosa and their antimicrobial susceptibility, as well as case reports, case studies, review articles, short communications, abstracts only, unpublished articles, and those with incomplete information. Additionally, only data from human-related studies were included, while data from environmental samples were excluded. All studies were screened against the inclusion and exclusion criteria from 13 May 2023, and data extracted from 5 October 2023.

Data extraction and quality control

All eligible studies were compiled and managed using EndNote 20. Duplicate articles were removed, and the remaining articles were systematically screened based on their titles and abstracts. Two authors, M.Z.S. and N.M.N.N.Z., independently reviewed the full texts of the articles to determine their eligibility and extracted the data based on predefined criteria. A third author, Z.Z.D., resolved any disagreements raised from data extraction. Ultimately, 40 eligible articles were selected and coded. Data from each article were then extracted into a Microsoft Excel table, which included essential information such as the title, author’s name, publication year, study region, study period, study design and methods, sample size, sample population, sample type, age group, gender, resistance patterns of the P. aeruginosa isolates, and the prevalence of MDR, XDR, ESBL- and MBL-producing P. aeruginosa. In this study, MDR was defined as resistance to at least one agent in at least 3 antibiotic classes, whereas XDR was defined as resistance to at least one agent in all but 1 or 2 antibiotic classes (Horcajada et al., 2019).

Data analysis

All 40 eligible studies were included in the meta-analysis. The analysis was conducted using metaprop codes in the meta (version 7.0-0) and metafor (version 4.4-0) packages of R (version 4.3.3; R Core Team, 2024), as implemented in RStudio (version 2023.12.1 Build 402) (Viechtbauer, 2010). The prevalence of resistant P. aeruginosa to each antibiotic, as well as the prevalence of MDR, XDR, ESBL- and MBL-producing P. aeruginosa isolates, along with 95% confidence intervals (CI), were calculated using the REML method for the random-effects model. Statistical heterogeneity among the studies was assessed using Cochran’s Q test and the inconsistency index (I2) (Huedo-Medina et al., 2006). An I2 value greater than 75% and a significance level below 0.05 (p-value) were considered indicative of significant heterogeneity. Publication bias was evaluated by examining a funnel plot, and significance was tested with Egger’s test only for groups of more than 10 studies.

Results

Selection and characteristics of the included studies

A PRISMA (Preferred Reporting Items for Systematic Reviews and Meta-Analyses) flow diagram illustrating our study selection process and literature search outcomes is presented in Fig. 1. We conducted an online search across three major databases—PubMed (n = 2,657), ScienceDirect (n = 3,374), and Scopus (n = 1,312)—which yielded a combined total of 7,343 records. After an initial eligibility check and the automatic removal of duplicate entries, 1,100 records remained for further screening based on their titles and abstracts. Through several rounds of manual review, 69 records were excluded, and 1,031 records proceeded to full-text evaluation. These records were meticulously assessed against predefined exclusion criteria (Fig. 1), resulting in the elimination of 991 records. Ultimately, 40 articles were deemed eligible and included in this qualitative analysis and meta-analysis.

The characteristics of the 40 included studies are detailed in Table S1. Studies were performed across 21 countries, with Iran reporting the highest number of P. aeruginosa infections (n = 614) in seven different studies over the last 5 years. Pakistan followed with 467 cases recorded in seven studies, while Italy, represented by a single study, reported 317 cases between 2018 and 2019 (Morroni et al., 2022). In total, 3,108 cases reported involving P. aeruginosa, employing antibiotic susceptibility testing with 48 different antimicrobials, were identified (Table S1). All 40 included studies were conducted between 2017 and 2022, published between 2019 and 2023 and were cross-sectional. P. aeruginosa isolates were collected from various age groups and characterized using either polymerase chain reaction (PCR) or the automated bacterial identification system VITEK 2, with a double-disk synergy test (DDST) utilized to detect ESBL and MBL genes. Antibiotic susceptibility tests against P. aeruginosa isolates were conducted using biochemical tests such as the Kirby-Bauer method (disk diffusion method) and broth microdilution. Among the 40 selected studies, 25 provided data on MDR, while only nine studies provided data on XDR. Data on ESBL- and MBL-producing P. aeruginosa were collected from 16 studies, respectively.

Five-year prevalence of P. aeruginosa infections

The estimated prevalence of P. aeruginosa infections among various age groups, as extrapolated from 26 studies, was found to be 22.9% (95% CI [14.4–31.4]) (Fig. 2). Notably, P. aeruginosa prevalence varied significantly across the region, ranging from as low as 2.6% to as high as 81.5%. Out of the initial pool of 40 studies, 14 were deemed ineligible due to insufficient data on the study population (Table S1). The observed asymmetrical distribution of effect estimates, depicted in the funnel plot of study distribution (Fig. 2), prompted a more granular analysis of the data based on subgroups. Stratifying the studies by region revealed notable differences, with the highest estimated P. aeruginosa prevalence observed in Asia at 24.7% (95% CI [13.5–36.0]), followed by Africa (18.9%, 95% CI [7.0–30.8]). Further stratification by country unveiled significant disparities. Nigeria, Iran, and Yemen showcased the highest prevalence estimates at 55.8% (95% CI [47.4–64.0]), 54.1% (95% CI [44.8–63.2]), and 49.0% (95% CI [41.9–56.1]), respectively, while Tanzania reported the lowest estimated prevalence of P. aeruginosa infections at 2.7% (95% CI [1.4–4.8]), followed by Thailand at 4.6% (95% CI [2.9–7.1]), all of which are represented by a single study (Table 1).

Figure 2 Forest and funnel plots depict the pooled prevalence of P. aeruginosa infections from 26 studies.

Fourteen studies, which did not report the total population numbers, have been excluded from this estimation. The pooled prevalence estimate was calculated using the random-effects model (top panel). The distribution of effect estimates is illustrated by a funnel plot (bottom panel). The figures were generated using RStudio software. Studies: (Abdelaziz et al., 2021; Abdeta et al., 2021; Ahmed et al., 2022; Alabdali, 2021; Ali et al., 2021, 2022; Bitew, Adane & Abdeta, 2023; Falodun, Musa & Oyelade, 2021; Jarjees, Jarjees & Qader, 2021; Kaluba et al., 2021; Kumari et al., 2022; Lee et al., 2019; Manyahi et al., 2020; Muhammad et al., 2020; Mustafai et al., 2023; Namaei et al., 2021; Nasser et al., 2020; Ngoi et al., 2021; Pandey, Mishra & Shrestha, 2021; Ruekit et al., 2022; Saleem & Bokhari, 2020; Shukla et al., 2021; Tilahun et al., 2022; Tran et al., 2022; Yong et al., 2021; Zahoor et al., 2023).

Table 1 The pooled prevalence of P. aeruginosa infections in different geographical regions.

Subgroup	Prevalence (%)
[95% CIs]	No. of studies	Sample size (P. aeruginosa isolates)	Sample population	I 2	p-value	
Regions	
Africa	18.9 [7.0–30.8]	8	343	2,234	98%	<0.01	
Asia	24.7 [13.5–36.0]	18	1,201	7,032	99%	<0.01	
Countries	
Egypt	29.9 [23.5–36.9]	1	58	194	NA	NA	
Ethiopia	8.0 [4.4–11.5]	3	89	1,008	75%	0.02	
India	17.2 [5.4–28.9]	2	126	603	91%	<0.01	
Iran	54.1 [44.8–63.2]	1	66	122	NA	NA	
Iraq	44.7 [0.0–100.0]	2	142	750	100%	<0.01	
Malaysia	19.6 [8.2–31.0]	2	87	505	87%	<0.01	
Nepal	18.6 [15.1–22.5]	1	84	452	NA	NA	
Nigeria	55.8 [47.4–64.0]	1	82	147	NA	NA	
Pakistan	21.3 [0.0–45.0]	6	305	2,531	99%	<0.01	
Somalia	19.2 [12.0–28.3]	1	19	99	NA	NA	
Taiwan	21.3 [19.0–23.7]	1	252	1,184	NA	NA	
Tanzania	2.7 [1.4–4.8]	1	11	402	NA	NA	
Thailand	4.6 [2.9–7.1]	1	20	431	NA	NA	
Vietnam	8.3 [5.2–12.4]	1	21	254	NA	NA	
Yemen	49.0 [41.9–56.1]	1	98	200	NA	NA	
Zambia	21.9 [17.8–26.3]	1	84	384	NA	NA	

Current antimicrobial resistance trends of P. aeruginosa

The antimicrobial susceptibility of P. aeruginosa isolates from the 40 included studies was assessed against a wide range of antibiotics. The prevalence estimates of resistant P. aeruginosa isolates tested against 48 different antibiotics are detailed in Table 2. Antibiotics were categorized into 14 groups, with the cephalosporin group being the most frequently utilized; specifically, ceftazidime and amikacin emerged as the most commonly employed antibiotics (37 studies each), while cefadroxil, cefpodoxime, erythromycin, amoxicillin, carbenicillin, ticarcillin, nalidixic acid, fusidic acid, and minocycline were the least frequently tested against P. aeruginosa (one study each). Our meta-analysis unveiled that resistant P. aeruginosa strains are prevalent across the majority of the antibiotics tested, albeit exhibiting variability (Table 2). Examination of antimicrobial resistance patterns for P. aeruginosa revealed high resistance rates, with 100.00% of isolates demonstrating resistance to cephalothin, cefazolin, cefpodoxime, erythromycin, ampicillin/sulbactam, ticarcillin, fusidic acid, and trimethoprim, followed by ampicillin (95.8%, 95% CI [90.2–100.0]) and ceftriaxone (93.1%, 95% CI [83.7–100.0]). Conversely, antimicrobials including polymyxin B (0.3%, 95% CI [0.0–1.3]) and polymyxin E (5.8%, 95% CI [1.5–10.2]), which are commonly used as the last resort for antimicrobial treatments, exhibited the lowest rates of P. aeruginosa resistance.

Table 2 The pooled prevalence of drug-resistant P. aeruginosa against 48 different antibiotics.

Antibiotics	Prevalence (%)
[95% CIs]	No. of resistant isolates	No. of studies	I 2	p-value	
1st gen. Cephalosporins	
Cephalothin (CEP)	100.0 [93.9–100.0]	27	2	0%	1	
Cefadroxil (CFR)	75.9 [62.8–86.1]	44	1	NA	NA	
Cefazolin (CFZ)	100.0 [97.0–100.0]	79	4	0%	1	
2nd gen. Cephalosporins	
Cefuroxime (CFX)	91.8 [82.7–100.0]	135	6	78%	<0.01	
Cefoxitin (FOX)	87.4 [71.2–100.00]	140	5	94%	<0.01	
3rd gen. Cephalosporins	
Ceftazidime (CAZ)	48.4 [40.7–56.2]	1,231	37	99%	0	
Ceftazidime + avibactam (CZA)	38.8 [27.1–51.5]	26	1	NA	NA	
Cefixime (CFM)	66.0 [27.9–100.0]	45	3	95%	<0.01	
Cefpodoxime (CPD)	100.0 [59.0–100.00]	7	1	NA	NA	
Ceftriaxone (CTR)	93.1 [83.7–100.0]	112	6	73%	<0.01	
Cefotaxime (CTX)	84.4 [70.0–98.7]	325	9	94%	<0.01	
4th gen. Cephalosporins	
Cefepime (CPM)	42.0 [31.8–52.1]	725	25	99%	0	
5th gen. Cephalosporins						
Ceftolozane + tazobactam (CXT)	20.5 [0.0–52.4]	40	2	97%	<0.01	
Aminoglycosides	
Amikacin (AMK)	28.5 [21.4–35.6]	775	37	97%	<0.01	
Gentamicin (GEN)	42.3 [34.0–50.6]	956	33	98%	<0.01	
Tobramycin (TOB)	41.3 [28.7–53.9]	484	17	98%	<0.01	
Carbapenems	
Doripenem (DOR)	45.1 [26.8–63.4]	136	4	92%	<0.01	
Ertapenem (ETP)	91.3 [71.0–100.0]	35	2	78%	0.03	
Imipenem (IMI)	31.9 [23.3–40.6]	669	31	98%	0	
Meropenem (MEM)	31.9 [24.1–39.7]	677	34	98%	0	
Macrolides	
Erythromycin (ERY)	100.0 [95.5–100.0]	80	1	NA	NA	
Monobactams	
Aztreonam (AZT)	43.3 [32.7–53.9]	605	20	96%	<0.01	
Penicillins	
Amoxicillin + clavulanic acid (AMC)	65.4 [44.2–86.7]	268	10	98%	<0.01	
Amoxicillin (AMX)	72.4 [59.1–83.3]	42	1	NA	NA	
Ampicillin (AMP)	95.8 [90.2–100.0]	204	6	70%	<0.01	
Ampicillin + sulbactam (SAM)	100.0 [95.5–100.0]	41	2	0%	1	
Carbenicillin (CAR)	54.1 [36.9–70.5]	20	1	NA	NA	
Piperacillin (PIP)	44.1 [27.2–61.0]	393	11	99%	0	
Piperacillin + tazobactam (TZP)	35.5 [28.0 – 43.1]	829	33	99%	0	
Ticarcillin (TIC)	100.0 [94.2–100.0]	62	1	NA	NA	
Ticarcillin + clavulanic acid (TCC)	73.2 [47.3–99.1]	216	5	99%	<0.01	
Phenicols	
Chloramphenicol (CHL)	84.4 [67.8–100.0]	138	3	93%	<0.01	
Quinolones	
Ciprofloxacin (CIP)	45.5 [37.9–53.2]	1,166	36	99%	0	
Levofloxacin (LEV)	44.5 [32.9–56.1]	531	19	98%	<0.01	
Nalidixic acid (NAL)	86.5 [79.3–91.9]	109	1	NA	NA	
Norfloxacin (NOR)	56.8 [30.8–82.7]	191	4	97%	<0.01	
Ofloxacin (OFX)	72.5 [64.1–80.9]	79	2	0%	0.98	
Tetracyclines	
Doxycycline (DOX)	66.8 [29.8–100.0]	123	3	98%	<0.01	
Minocycline (MIN)	59.1 [46.3–71.0]	39	1	NA	NA	
Tetracycline (TET)	71.9 [38.0–100.00]	101	6	100%	<0.01	
Tigecycline (TGC)	54.3 [18.4–90.1]	140	5	99%	<0.01	
Others	
Colistin/Polymyxin E (COL)	5.8 [1.5–10.2]	73	13	85%	<0.01	
Fosfomycin (FOS)	33.3 [25.4–41.1]	46	2	0%	0.54	
Fusidic acid (FUS)	100.0 [89.1–100.0]	32	1	NA	NA	
Nitrofurantoin (NIT)	92.4 [78.1–100.0]	127	3	89%	<0.01	
Polymyxin B (POL)	0.3 [0.0–1.3]	2	4	0%	0.65	
Sulfamethoxazole + trimethoprim (SXT)	74.7 [50.3–99.0]	220	8	97%	<0.01	
Trimethoprim (TMP)	100.0 [98.4–100.0]	102	2	0%	1	

Further stratification by class of antibiotics revealed substantial disparities—even among the same antibiotic class. Analysis of antimicrobial resistance patterns of β-lactam antibiotics in P. aeruginosa indicated widespread resistance, with all isolates displaying resistance to cephalothin and cefazolin, both belonging to the first-generation cephalosporins. Additionally, 100.0% of P. aeruginosa isolates also demonstrating resistance to cefpodoxime, a third-generation cephalosporin (Fig. 3). Among the 13 cephalosporin antibiotics tested, ceftolozane in combination with tazobactam, a β-lactamase inhibitor recorded the lowest rate of P. aeruginosa resistance (20.5%, 95% CI [0.0–52.4]), followed by ceftazidime/avibactam (38.8%, 95% CI [27.1–51.5]) and cefepime (42.0%, 95% CI [31.8–52.1]). By popularity, ceftazidime appeared to be the most commonly employed cephalosporins, used in 37 studies, while cefadroxil, ceftazidime/avibactam, and cefpodoxime were the least commonly used cephalosporins against P. aeruginosa. Out of the nine penicillins examined, ticarcillin and ampicillin exhibited the highest resistance rates, with 100.0% and 95.8% of the P. aeruginosa isolates showing resistance, respectively, while piperacillin in combination with tazobactam revealed the lowest rate of P. aeruginosa resistance (35.5%, 95% CI [28.0–43.1]). Among the four carbapenems, the highest resistance was recorded in ertapenem (91.3%, 95% CI [71.0–100.0]), while imipenem and meropenem exhibited resistance in 31.9% of P. aeruginosa isolates.

Figure 3 Antimicrobial resistance patterns of three different classes of β-lactam antibiotics.

In P. aeruginosa: cephalosporins (A), penicillins (B) and carbapenems (C).

Recent prevalence of multidrug-resistant P. aeruginosa

The prevalence estimates of multidrug-resistant (MDR) P. aeruginosa was notably high across various studies, with 44% (11 out of 25) of the included studies reporting multiple resistance rates exceeding 50.0% for total P. aeruginosa isolates (Fig. 4). Our meta-analysis indicated that the combined prevalence of MDR P. aeruginosa was calculated to be 46.0% (95% CI [37.1–55.0]), with significant heterogeneity (I2 = 98%, τ2 = 0.0485, p < 0.01). The maximum and minimum prevalence rates of MDR P. aeruginosa were calculated to be 100.0% (Ali et al., 2022) and 11.2% (Jayakumar et al., 2020), respectively. The prevalence of MDR was estimated on P. aeruginosa isolates from 1,922 clinical samples collected in 25 studies within the past 5 years. Non-publication bias, evident from a symmetrical funnel plot (Fig. 4), was statistically confirmed using Egger’s test (p = 0.1014).

Figure 4 Forest and funnel plots representing the pooled prevalence of multidrug-resistant (MDR) P. aeruginosa in the past 5 years.

The prevalence was estimated by pooling 25 selected studies using the random-effects model (top panel). The effect estimate distributions are shown in a funnel plot (bottom panel). Figures were generated using RStudio software. Studies: (Abavisani et al., 2021; Abdelaziz et al., 2021; Abdeta et al., 2021; Ali et al., 2022; Alnimr & Alamri, 2020; Bitew, Adane & Abdeta, 2023; Dehbashi et al., 2020; El-Mahdy & El-Kannishy, 2019; Falodun, Musa & Oyelade, 2021; Jarjees, Jarjees & Qader, 2021; Jayakumar et al., 2020; Karampoor et al., 2022; Khademi et al., 2021; Liu et al., 2020; Mohamed et al., 2022; Muhammad et al., 2020; Namaei et al., 2021; Nasser et al., 2020; Ngoi et al., 2021; Pandey, Mishra & Shrestha, 2021; Park & Koo, 2022; Saleem & Bokhari, 2020; Shukla et al., 2021; Tilahun et al., 2022; Zahoor et al., 2023).

When stratified by different regions, the highest prevalence of MDR P. aeruginosa was observed in Africa, estimated at 50.9% (95% CI [33.5–68.2]), followed by Asia (44.3%, 95% CI [33.7–55.0]) (Table 3). In total, 820 MDR P. aeruginosa isolates were documented over 5 years (2018–2023), in which 645 MDR isolates were reported in Asia. Furthermore, our meta-analysis indicated that Iraq recorded the highest levels of MDR P. aeruginosa prevalence (74.2%, 95% CI [61.5–84.5]), followed by Yemen (66.3%, 95% CI [56.1–75.6]) and Egypt (64.3%, 95% CI [47.4–81.3]), while Nepal reported the lowest prevalence (14.3%, 95% CI [7.6–23.6]). Iran, in addition to having an MDR P. aeruginosa prevalence of 44.8% (95% CI [36.5–53.1]), documented the highest number of MDR P. aeruginosa isolates in the past 5 years (n = 199), followed by Pakistan (n = 119) with an MDR prevalence of 53.7% (95% CI [20.4–86.9]) (Table 3).

Table 3 The recent pooled prevalence of MDR P. aeruginosa in different geographical regions.

Subgroup	Prevalence (%)
[95% CIs]	No. of MDR isolates	Total P. aeruginosa isolates	No. of studies	I 2	p-value	
Regions	
Africa	50.9 [33.5–68.2]	175	328	7	93%	<0.01	
Asia	44.3 [33.7–55.0]	645	1,594	18	99%	<0.01	
Countries	
China	23.0 [15.6–31.9]	26	113	1	NA	NA	
Egypt	64.3 [47.4–81.3]	90	138	2	77%	0.04	
Ethiopia	48.5 [10.0–87.1]	51	89	3	95%	<0.01	
India	31.5 [0.0–71.7]	75	249	2	98%	<0.01	
Iran	44.8 [36.5–53.1]	199	444	5	69%	0.01	
Iraq	74.2 [61.5–84.5]	46	62	1	NA	NA	
Malaysia	20.4 [10.6–33.5]	11	54	1	NA	NA	
Nepal	14.3 [7.6–23.6]	12	84	1	NA	NA	
Nigeria	30.5 [20.8–41.6]	25	82	1	NA	NA	
Pakistan	53.7 [20.4–86.9]	119	299	4	99%	<0.01	
Saudi Arabia	56.7 [44.0–68.8]	38	67	1	NA	NA	
Somalia	47.4 [24.4–71.1]	9	19	1	NA	NA	
South Korea	43.5 [34.7–52.7]	54	124	1	NA	NA	
Yemen	66.3 [56.1–75.6]	65	98	1	NA	NA	

Moreover, the prevalence of extensively drug-resistant (XDR) P. aeruginosa was markedly high, with 19.6% (95% CI [4.3–34.9]) of the total P. aeruginosa isolates (n = 139) being resistant to most available antibiotics (Fig. 5). The highest and lowest prevalence rates of XDR P. aeruginosa were calculated to be 72.5% (95% CI [61.4–81.9]) (El-Mahdy & El-Kannishy, 2019) and 2.8% (95% CI [0.1–14.5]) (Abdeta et al., 2021), respectively with significant heterogeneity (I2 = 96%, τ2 = 0.0533, p < 0.01).

Figure 5 Forest and funnel plots representing the recent pooled prevalence of extensively drug-resistant (XDR) P. aeruginosa.

The estimate of prevalence was calculated by pooling 9 selected studies using the random-effects model (top panel). The distribution of effect estimates is shown by a funnel plot (bottom panel). Figures were generated using RStudio software. Studies: (Abavisani et al., 2021; Abdeta et al., 2021; El-Mahdy & El-Kannishy, 2019; Falodun, Musa & Oyelade, 2021; Jarjees, Jarjees & Qader, 2021; Namaei et al., 2021; Pandey, Mishra & Shrestha, 2021; Saleem & Bokhari, 2020; Shukla et al., 2021).

Five-year patterns of extended-spectrum β-lactamase-producing P. aeruginosa

In our meta-analysis, out of 40 eligible studies, only 16 studies with a combined sample size of 371, reported ESBL-producing P. aeruginosa in patients (Fig. 6). The overall prevalence of ESBL-producing P. aeruginosa was found to be 33.4% (95% CI [23.6–43.2]), with substantial heterogeneity between studies (I2 = 95%, τ2 = 0.0346, p < 0.01). The presence of publication bias, indicated by an asymmetrical funnel plot, was statistically confirmed using Egger’s test (p = 0.0126). Our analysis indicated that the highest prevalence of ESBL-producing P. aeruginosa was observed in a study from Iran (66.0%, 95% CI [55.8–75.2]) (Shalmashi et al., 2022), while the lowest prevalence was reported in Thailand (5.0%, 95% CI [0.1–24.9]) (Ruekit et al., 2022). Regionally, Africa had the highest prevalence of ESBL-producing P. aeruginosa at 35.0% (95% CI [24.6–45.4]), followed by Asia at 34.2% (95% CI [21.8–46.6]), and Europe at 20.0% (95% CI [4.3–48.1]) (Table 4). When stratified by countries, the highest prevalence of ESBL-producing P. aeruginosa was observed in Yemen, estimated at 56.9% (95% CI [44.0–69.2]), followed by India (51.3%, 95% CI [41.8–60.7]), and Iran (43.5%, 95% CI [14.4–72.5]).

Figure 6 Forest and funnel plots representing the current pooled prevalence of ESBL-producing P. aeruginosa.

The estimate of prevalence was calculated by pooling 16 selected studies using the random-effects model (top panel). The distribution of effect estimates, shown by a funnel plot (bottom panel). Figures were generated using RStudio software. Studies: (Abavisani et al., 2021; Ali et al., 2021, 2022; Bitew, Adane & Abdeta, 2023; Dehbashi et al., 2020; Falodun, Musa & Oyelade, 2021; Jarjees, Jarjees & Qader, 2021; Morroni et al., 2022; Muhammad et al., 2020; Mustafai et al., 2023; Nasser et al., 2020; Pandey, Mishra & Shrestha, 2021; Ruekit et al., 2022; Shalmashi et al., 2022; Shukla et al., 2021; Tilahun et al., 2022).

Table 4 The pooled prevalence of ESBL-producing P. aeruginosa in different geographical regions.

Subgroup	Prevalence (%)
[95% CIs]	No. of ESBL-producing isolates	Total P. aeruginosa isolates	No. of studies	I 2	p-value	
Regions	
Africa	35.0 [24.6–45.4]	47	135	3	11%	0.32	
Asia	34.2 [21.8–46.6]	321	939	12	97%	<0.01	
Europe	20.0 [4.3–48.1]	3	15	1	NA	NA	
Countries	
Ethiopia	41.2 [28.0–54.3]	22	53	2	0%	0.42	
India	51.3 [41.8–60.7]	59	115	1	NA	NA	
Iran	43.5 [14.4–72.5]	120	258	3	97%	<0.01	
Iraq	27.4 [16.9–40.2]	17	62	1	NA	NA	
Italy	20.0 [4.3–48.1]	3	15	1	NA	NA	
Nepal	10.7 [5.0–19.4]	9	84	1	NA	NA	
Nigeria	30.5 [20.8–41.6]	25	82	1	NA	NA	
Pakistan	32.0 [11.5–52.4]	78	335	4	96%	<0.01	
Thailand	5.0 [0.1–24.9]	1	20	1	NA	NA	
Yemen	56.9 [44.0–69.2]	37	65	1	NA	NA	

Trends of metallo-β-lactamase-producing P. aeruginosa in the past 5 years

Similarly, out of the 40 eligible studies included in our meta-analysis, only 16 studies, encompassing a total sample size of 201, reported on metallo-β-lactamase (MBL)-producing P. aeruginosa in the past 5 years (Fig. 7). After pooling the results of these studies, the prevalence of MBL-producing P. aeruginosa was estimated to be 16.0% (95% CI [9.8–22.3]), with significant heterogeneity between studies (I2 = 93%, τ2 = 0.0143, p < 0.01). Publication bias was statistically confirmed through Egger’s test (p = 0.0001), as indicated by an asymmetrical funnel plot. When stratified according to different regions, the highest prevalence of MBL-producing P. aeruginosa was estimated to be 33.3% (95% CI [11.8–61.6]), which was calculated for Europe from a single study, followed by the estimates for Asia (17.7%, 95% CI [11.2–24.2]) and Africa (0.2%, 95% CI [0.0–1.8]) (Table 5). Our meta-analysis further revealed that the highest MBL-producing P. aeruginosa prevalence estimates were recorded in Italy (33.3%, 95% CI [11.8–61.6]), followed by Yemen (32.3%, 95% CI [21.2–45.1]) and India (19.9%, 95% CI [14.9–24.8]), while the lowest prevalence was recorded in Nigeria (0.0%, 95% CI [0.0–4.4]).

Figure 7 Forest and funnel plots representing the pooled prevalence of metallo-β-lactamase (MBL)-producing P. aeruginosa in the past 5 years.

The estimate of prevalence was calculated by pooling 16 studies using the random-effects model (top panel). The distribution of effect estimates, showed by a funnel plot (bottom panel). Figures were generated using RStudio software. Studies: (Abdeta et al., 2021; Jayakumar et al., 2020; Shukla et al., 2021; Abavisani et al., 2021; Dehbashi et al., 2020; Karampoor et al., 2022; Namaei et al., 2021; Rad et al., 2021; Morroni et al., 2022; Ngoi et al., 2021; Pandey, Mishra & Shrestha, 2021; Falodun, Musa & Oyelade, 2021; Ali et al., 2021, 2022; Saleem & Bokhari, 2020; Nasser et al., 2020).

Table 5 The pooled prevalence of MBL-producing P. aeruginosa in different geographical regions.

Subgroup	Prevalence (%)
[95% CIs]	No. of MBL-producing isolates	Total P. aeruginosa isolates	No. of studies	I 2	p-value	
Regions	
Africa	0.2 [0.0–1.8]	1	118	2	0%	0.33	
Asia	17.7 [11.2–24.2]	195	1114	13	84%	<0.01	
Europe	33.3 [11.8–61.6]	5	15	1	NA	NA	
Countries	
Ethiopia	2.8 [0.1–14.5]	1	36	1	NA	NA	
India	19.9 [14.9–24.8]	50	249	2	0%	0.36	
Iran	18.0 [1.1–34.9]	71	411	5	93%	<0.01	
Italy	33.3 [11.8–61.6]	5	15	1	NA	NA	
Malaysia	16.7 [7.9–29.3]	9	54	1	NA	NA	
Nepal	8.3 [3.4–16.4]	7	84	1	NA	NA	
Nigeria	0.0 [0.0–4.4]	0	82	1	NA	NA	
Pakistan	14.7 [9.1–20.4]	37	251	3	25%	0.26	
Yemen	32.3 [21.2–45.1]	21	65	1	NA	NA	

Discussion

P. aeruginosa continues to remain a leading cause of high morbidity and mortality globally, renowned for its capacity to give rise to severe infections—hospital-acquired infections such as ventilator-associated pneumonia and various sepsis syndromes, particularly in individuals with compromised immune systems, those with chronic illnesses, and individuals with CF (Malhotra, Hayes & Wozniak, 2019). This pathogen poses a substantial threat to public health due to its multidrug resistance, associated with intrinsically advanced antibiotic resistance mechanisms. In a recent global disease burden report, P. aeruginosa was listed as one of the six pathogens, each responsible for over 250,000 deaths linked to antimicrobial resistance. These pathogens, listed in descending order of the number of deaths, were Escherichia coli, Staphylococcus aureus, Klebsiella pneumoniae, Streptococcus pneumoniae, Acinetobacter baumannii, and P. aeruginosa (Murray et al., 2022). Collectively, these six pathogens accounted for 3.57 million out of the 4.95 million deaths associated with antimicrobial resistance worldwide in 2019. However, the worldwide rise of antimicrobial-resistant P. aeruginosa strains, which restrict the selection of effective treatments for pneumonia, has emerged as the primary challenge in treating P. aeruginosa infections. Recently, the World Health Organization (WHO) identified carbapenem-resistant P. aeruginosa as one of the top three bacterial species requiring new antibiotics for effective treatment. Furthermore, the overuse of antibiotics during treatment speeds up the emergence of multidrug-resistant P. aeruginosa strains, rendering empirical antibiotic therapies including aminoglycosides, quinolones and β-lactams ineffective against this pathogen (Pang et al., 2019). Therefore, assessing the P. aeruginosa resistance in a population is crucial for developing targeted therapeutic strategies to mitigate the incidence of mortality and morbidity caused by P. aeruginosa infections. This is especially crucial for patients with CF, who often experience chronic pseudomonal colonization and recurrent infections, leading to disease exacerbations partly due to the limited availability of effective antipseudomonal therapeutics. This necessitates comprehensive data on the recent prevalence and patterns of antimicrobial-resistant P. aeruginosa worldwide. However, to our knowledge, such data is currently unavailable.

In this SRMA, we present the prevalence of multidrug-resistant and beta-lactamase-producing P. aeruginosa, by combining all eligible data on the prevalence of antimicrobial-resistant P. aeruginosa from community- and hospital-acquired infections, as reported in the 40 selected studies in the past 5 years (Table S1). As anticipated from studies conducted using different methods in various backgrounds and settings, our findings exhibited considerable heterogeneity. This significant variability was also noted in our previous research on the prevalence of multidrug-resistant diarrheagenic E. coli and Shigella spp. (Salleh et al., 2022a, 2022b). The heterogeneity is likely attributed to differences in methodologies, sample sizes, and research settings, including study regions, study periods, and population ages across different studies. This was expected, given that our SRMA employed a random-effects model, which assumes heterogeneity, unlike a fixed-effects model (Imrey, 2020). Nevertheless, to the best of our knowledge, our SRMA is the first to assess the current prevalence of multidrug-resistant and beta-lactamase-producing P. aeruginosa in Asia and Africa, and we hope it will be valuable for designing targeted strategies for treating pseudomonal infections.

The estimated prevalence of pseudomonal infections across various age groups from different settings was 22.9% (95% CI [14.4–31.4]) (Fig. 2). Notably, the prevalence of P. aeruginosa showed significant regional variation, with infection rates ranging from 2.6% to 81.5%. For comparison, the prevalence of community-acquired pseudomonal infections in 2015 was estimated to be 11.3% (Restrepo et al., 2018), significantly lower than the present study. This is expected, as our study included eligible data on the prevalence of P. aeruginosa from community-and hospital-acquired infections in the past 5 years. However, it is important to note that not all studies included in this SRMA specified whether the infections were community-or hospital-acquired. Many studies did not provide this distinction and, in some cases, combined their findings from various settings. As such, the data in our analysis includes both community-and hospital-acquired infections where available, but the exact categorization was not consistently specified across all included studies.

The majority of studies on the prevalence of P. aeruginosa included in this SRMA were conducted in Pakistan, representing six studies with a total of 305 confirmed cases in recent years. These studies focused on P. aeruginosa isolates obtained from a variety of clinical specimens, including sputum, pus, urine, and blood, sourced from patients diagnosed with different types of infections, such as urinary tract infections, bloodstream infections, wound infections, and respiratory infections. The prevalence of P. aeruginosa in Pakistan was calculated at 21.3% (95% CI [0.0–45.0]), slightly lower than our pooled estimate for pseudomonal infections. Nevertheless, the prevalence of P. aeruginosa calculated in our SRMA was notably higher than the prevalence estimate in a study from Pakistan in 2019, which reported an estimate of 9.3% (Farooq et al., 2019). The estimate of P. aeruginosa infections in Nigeria was 55.8% (95% CI [47.4–64.0])—significantly higher than the prevalence of P. aeruginosa in general (Table 1). This concentration is not surprising, as P. aeruginosa has been a significant concern in sub-Saharan Africa for the past few decades. Sub-Saharan Africa carries a substantial burden of infectious diseases, potentially related to high poverty levels and inadequate water, sanitation, and hygiene practices. Furthermore, sub-Saharan Africa experiences the highest prevalence of deaths associated with antimicrobial resistance worldwide (Arowolo et al., 2023). In Nigeria, P. aeruginosa has been reported in both clinical and environmental settings, reflecting its widespread prevalence. For instance, a study reported a prevalence of 50% in ear infections (Adedeji, Fagade & Oyelade, 2007), which indicates that it was a predominant pathogen in such cases. Additionally, a more recent study found that 66.7% of P. aeruginosa isolates were recovered from sewage samples (Adesoji et al., 2023). This high percentage emphasizes the significant prevalence of P. aeruginosa in environmental reservoirs in Nigeria, particularly wastewater, underpinning the role of sewage as a potential source of P. aeruginosa contamination with serious public health implications. Interestingly, the prevalence of P. aeruginosa infections in Tanzania was the lowest among all countries studied in our SRMA (2.7%, 95% CI [1.4–4.8]). This was slightly lower than the prevalence of P. aeruginosa isolated from diabetic foot ulcers in Tanzania recently, recorded at 4.8% (95% CI [1.9–9.6]) (Makeri et al., 2023). Similarly, it was estimated that the African continental prevalence of P. aeruginosa at 11.8% (95% CI [8.7–15.2]) (Makeri et al., 2023), slightly lower than our data from Africa (18.9%, 95% CI [7.0–30.8]) (Table 1). Nevertheless, P. aeruginosa has been regarded as one of the two dominant species in diabetic foot ulcers in Africa – the other being S. aureus (Macdonald et al., 2021).

Pseudomonal infections are becoming increasingly resistant to certain antibiotics, and the organism may develop resistance during treatment. To combat this, especially in high-risk cases such as severe sepsis, septicemia, and inpatient neutropenia, it is recommended to use two agents from different antibiotic classes (Qureshi, 2023). A common treatment for P. aeruginosa infection includes a combination of antipseudomonal β-lactams such as penicillin or cephalosporin, and an aminoglycoside. Alternatively, carbapenems like imipenem or meropenem can be used with quinolones and an aminoglycoside. Except in febrile neutropenic patients, where monotherapy with ceftazidime or a carbapenem is advised, a two-drug regimen is always generally recommended to treat P. aeruginosa infections (Qureshi, 2023). When P. aeruginosa isolates are susceptible to both traditional non-carbapenem β-lactam agents (such as piperacillin-tazobactam, ceftazidime, cefepime, or aztreonam) and carbapenems, it is recommended to use the traditional β-lactam agents over carbapenems to preserve the activity of carbapenems for future. For infections caused by P. aeruginosa isolates that are resistant to any carbapenems but are susceptible to traditional β-lactams, it is suggested to administer the traditional agents using high-dose extended-infusion therapy (Tamma et al., 2023). In this meta-analysis, resistance was common among P. aeruginosa isolates across the majority of the antibiotics tested, albeit exhibiting variability (Table 2). For instance, the organism recorded high resistance rates against first-generation cephalosporins. The prevalence of P. aeruginosa resistant against cephalothin and cefazolin was the highest, both estimated at 100.0%, followed by cefadroxil at 75.9%. Other cephalosporins including cefuroxime, cefoxitin, cefpodoxime, ceftriaxone, and cefotaxime also recorded high resistance rates ranging from 84.4% to 100.0%. Furthermore, high resistance rates were observed in the penicillin class of antibiotics, with over 40% of P. aeruginosa isolates showing resistance to piperacillin (44.1%, 95% CI [27.2–61.0]), amoxicillin (72.4%, 95% CI [59.1–83.3]), ampicillin (95.8%, 95% CI [90.2–100.0]), and ticarcillin (100.0%, 95% CI [94.2–100.0]). Although resistance to penicillins was relatively high among P. aeruginosa isolates, combining these antibiotics with β-lactamase inhibitors such as clavulanic acid and tazobactam reduced resistance to amoxicillin, piperacillin, and ticarcillin. For example, combining amoxicillin and ticarcillin with clavulanic acid decreased the prevalence of resistant P. aeruginosa from 72.4% to 65.4% (95% CI [44.2–86.7]) and from 100.0% to 73.2% (95% CI [47.3–99.1]), respectively. Similarly, combining piperacillin with tazobactam reduced P. aeruginosa resistance from 44.1% to 35.5% (95% CI [28.0–43.1]). In essence, β-lactamase inhibitors enhance the effectiveness of β-lactam antibiotics against resistant strains of P. aeruginosa by shielding the antibiotics from β-lactamase degradation, thereby increasing susceptibility rates. High resistance rates to almost all classes of antimicrobials are concerning and may indicate the excessive and unjustified use of antibiotics in treating pseudomonal infections in general healthcare. Adjustment to their treatment protocols by tailoring antibiotic choices to regional resistance patterns, emphasizing the use of β-lactamase inhibitors in areas with high resistance, and strengthening antimicrobial stewardship programs to reduce excessive antibiotic use is crucial.

Furthermore, the significant resistance to many first-line drugs has led to relatively high rates of MDR in P. aeruginosa isolates reported in our SRMA in the past 5 years. Evidence from this study indicates that multidrug resistance poses a significant challenge in treating pseudomonal infections, with nearly half of P. aeruginosa strains now showing resistance to multiple antibiotics. Our meta-analysis revealed that the current prevalence of MDR P. aeruginosa was estimated at 46.0% (95% CI [37.1–55.0]) (Fig. 4), slightly higher than previously reported at 38.3% of MDR P. aeruginosa bloodstream infections in 2020 (Recio et al., 2020) and substantially higher than the report from a 2019–2020 single-center retrospective case control study in the USA recorded at 29.8% (Yang et al., 2023). Most of the studies included in the current analysis reported a high prevalence of MDR strains, in which the prevalence of MDR P. aeruginosa ranged from a minimum of 11.2% to a maximum of 100.0% (Fig. 4). This is worrying, as pseudomonal resistance to multiple antibiotics has been steadily increasing among the global population over the past few decades, particularly against aminoglycosides, quinolones and β-lactams, rendering empirical antibiotic therapies ineffective against the pathogen (Pang et al., 2019). Moreover, in recent years, there has been a rising prevalence of XDR P. aeruginosa strains, with rates ranging between 15% and 30% in certain geographical regions—the majority of European countries reported resistance rates exceeding 10% for all antimicrobial groups under surveillance (Horcajada et al., 2019). Our analysis revealed that the current prevalence of XDR P. aeruginosa was markedly high, with an estimation at 19.6% (95% CI [4.3–34.9]) of the total P. aeruginosa isolates being resistant to at least one agent in all but 1 or 2 antibiotic classes (Fig. 5). The rise in bacterial resistance to multiple antibiotics, coupled with the scarcity of new drugs in development, has become a significant clinical and public health issue globally, especially in the case of MDR and XDR and P. aeruginosa. While novel agents like ceftolozane-tazobactam and ceftazidime-avibactam have expanded treatment options, polymyxins often remain the only effective option in certain cases (Horcajada et al., 2019). Our analysis showed that polymyxin B (0.3%, 95% CI [0.0–1.3]) and colistin/polymyxin E (5.8%, 95% CI [1.5–10.2]), commonly used as last-resort antimicrobial treatments, had the lowest rates of P. aeruginosa resistance. Although relatively low, resistance to polymyxins can develop when used as monotherapy (Samal et al., 2021). Alas, due to nephrotoxicity and neurotoxicity being the primary dose-limiting factors for polymyxin monotherapy, increasing the dose to prevent resistance is viewed as a non-feasible solution, which restrict their safe use, particularly in patients with pre-existing kidney conditions or those requiring prolonged treatment (Özkarakaş et al., 2023; Pogue, Ortwine & Kaye, 2016). Consequently, innovative strategies are necessary to preserve this last-resort class of antibiotics. A previous study has suggested the use of polymyxin B in combination with enrofloxacin to treat XDR P. aeruginosa infections. The study showed the combination was synergistic against the pathogen, with ≥2 to 4 log10 kill at 24 h in the static time-kill studies, underlying the utmost importance of a rational designed combination therapy to treat XDR superbugs (Lin et al., 2018).

The high prevalence of MDR and XDR P. aeruginosa continues to pose significant health challenges globally. The acquisition of ESBL and MBL genes is one of several mechanisms that can contribute to an increase in MDR and XDR strains by enabling the hydrolysis and inactivation of a wide range of β-lactam antibiotics, including penicillins, cephalosporins, and carbapenems (Elfadadny et al., 2024). ESBL genes, often located on plasmids, facilitate the rapid spread of resistance through horizontal gene transfer, leading to multi-drug resistance when co-located with other resistance genes. MBLs, with their broad spectrum of activity against β-lactams and resistance to traditional β-lactamase inhibitors, further complicate treatment options. Both ESBL and MBL genes are highly mobile, found on mobile genetic elements such as integrons, plasmids, transposons, insertion sequences, as well as bacteriophages, promoting their dissemination across bacterial species and strains, horizontally by conjugation, transformation or, in the case of bacteriophages, by transduction (Castanheira, Simner & Bradford, 2021; Michaelis & Grohmann, 2023). In our meta-analysis, the prevalence of ESBL-producing P. aeruginosa was 33.4% (95% CI [23.6–43.2]), while the prevalence of MBL-producing strains was found to be 16.0% (95% CI [9.8–22.3]), highlighting the impact of factors such as antibiotic misuse, inadequate infection control practices, and regional variability in resistance genes, thus emphasizing the need for enhanced stewardship, diagnostics, and novel treatments. The highest prevalence of ESBL-producing P. aeruginosa was recorded in Yemen (56.9%), followed by India (51.3%) and Iran (43.5%) (Table 4), while the highest prevalence of MBL-producing P. aeruginosa was recorded in Italy (33.3%), followed by Yemen (32.3%) and India (19.9%) (Table 5). Carbapenem-resistant P. aeruginosa harbors several ESBLs, including those of Ambler class A, which encompasses various enzymes like Pseudomonas extended resistance bla. In a recent report from Iraq, blaOXA-10 and blaPER-1 were the most prevalent, found in 59.3% and 44.4% of cases, respectively, while blaSHV was the least common, with an abundance of 11.1% among the ESBL-producing isolates collected from burn patients (Polse, Khalid & Mero, 2023). The study showed that isolates with the blaOXA-10 gene were fully resistant to piperacillin, cefepime, and ceftazidime, and demonstrated high to moderate resistance to other tested antibiotics. Isolates harboring the blaPER-1 gene exhibited complete resistance to piperacillin, ceftazidime, and meropenem, and also showed high to moderate resistance to other antibiotics, whereas isolates that carried the blaSHV gene were almost entirely resistant to all antibiotics tested (Polse, Khalid & Mero, 2023). Additionally, there are several types of Amber class B MBLs present in carbapenem-resistant P. aeruginosa, including imipenemase (IMP), Verona integron-encoded metallo-β-lactamase (VIM), São Paulo metallo-β-lactamase (SPM), Germany imipenemase (GIM), New Delhi metallo-β-lactamase (NDM), and Florence imipenemase (FIM) (Hong et al., 2015). In our meta-analysis, ertapenem showed the highest resistance rate at 91.3% out of the four carbapenems, followed by doripenem at 45.1%. Imipenem and meropenem exhibited resistance in 31.9% of P. aeruginosa isolates (Table 2). In comparison, China recorded resistance rates at 43.4% and 40.9% in P. aeruginosa isolated from 2003 to 2011 against imipenem and meropenem, respectively (Xu et al., 2013), while Latin American countries reported resistance rates of up to 66% for P. aeruginosa isolated between 2002 and 2013 (Labarca et al., 2016). Nevertheless, high resistance against ertapenem was also recorded in Iran (74.4%) (Sheikh et al., 2014), suggesting its inferior efficacy compared to other carbapenems due to its poor ability to penetrate the outer membrane of the bacterium.

Our SRMA offers a current comprehensive analysis of antimicrobial-resistant P. aeruginosa from the past 5 years. However, it has several noteworthy limitations. Firstly, despite including a substantial number of studies (n = 40), not all countries across the globe were represented. Consequently, the current estimated prevalence may not fully reflect the true extent of antimicrobial-resistant P. aeruginosa globally. Nonetheless, data from 21 countries were collected, encompassing a substantial number of pseudomonal cases (n = 3,108) with 48 different antibiotics tested (Table S1). Secondly, our SRMA revealed significant heterogeneity, which is typical in meta-analyses of prevalence estimation. This was anticipated, as we employed a random-effects model that inherently accounts for heterogeneity (Imrey, 2020). Significant heterogeneity often reflects variations in healthcare infrastructure, antibiotic use policies, socio-economic factors, and local pathogen prevalence. Random-effects model was chosen because it assumes that the true effect size varies between studies rather than being fixed, which aligns with the significant heterogeneity observed in our dataset. The model provides a more robust and generalizable result despite the observed variability. This heterogeneity likely arises from differences in study designs, populations, geographical regions, or detection methods. A more granular regional meta-analysis could provide tailored data that highlights specific drivers of AMR in a particular area, which is useful for local intervention strategies. Thirdly, we were unable to account for the effects of age and gender distribution on the prevalence of antimicrobial-resistant P. aeruginosa due to the variability in data reporting among the included studies. While some studies provided antimicrobial resistance data from various age groups, many did not specify patient age and gender groups at all. Therefore, the impact of age and gender distribution on P. aeruginosa prevalence could not be assessed due to these inconsistencies in data reporting. Nevertheless, we believe that our SRMA presents vital information on the current prevalence of multidrug-resistant and β-lactamase-producing P. aeruginosa that would be helpful to researchers, clinicians, and governments.

Conclusion

Our SRMA provides substantial evidence of the current distribution of antimicrobial-resistant P. aeruginosa over the past 5 years. The prevalence estimates indicated a significantly high proportion of MDR P. aeruginosa, representing a major public health burden that continues to pose significant health challenges globally, including increased mortality, higher healthcare costs, and the spread of resistant infections. These issues strain healthcare systems, especially in low- and middle-income countries, and threaten medical advances, leading to health inequities due to unequal access to effective treatment. These issues further challenge global efforts to combat AMR, such as the WHO’s Global Action Plan, which emphasizes surveillance, rational antibiotic use, and research investment. Our meta-analysis estimated that the prevalence of MDR and XDR P. aeruginosa in the last 5 years was 46.0% and 19.6%, respectively, while the prevalence of ESBL-and MBL-producing P. aeruginosa strains was 33.4% and 16.0%, respectively. High prevalence rates reflect the influence of factors such as antibiotic misuse, insufficient infection control measures, and regional variations in resistance mechanisms, highlighting the critical need for improved antimicrobial stewardship, rapid diagnostics, and the development of novel therapeutic options. Specifically, our prevalence data underscores the critical need for routine microbiological testing to identify ESBL- and MBL-producing P. aeruginosa, enabling clinicians to choose more targeted and effective therapies while avoiding the unnecessary use of broad-spectrum antibiotics. Although the prevalence varied across different countries, the findings suggest that multidrug resistance is a critical public health threat that requires urgent and appropriate interventions. Continuous monitoring of antimicrobial-resistant P. aeruginosa through rigorous drug susceptibility tests is essential. Implementing reliable and effective antibiotic resistance mitigation strategies could lead to better outcomes for the treatment and control of pseudomonal infections across the globe.

Supplemental Information

Supplemental Information 1 Characteristics of the 40 included studies, performed across 21 countries in this systematic review and meta-analysis (SRMA).

ND, no data; KB, Kirby-Bauer disk diffusion; DDST, double disk synergy test; PCR, polymerase chain reaction; AMC, amoxicillin and clavulanic acid; AMK, amikacin; AMP, ampicillin; AMX, amoxicillin; AZT, aztreonam; CAZ, ceftazidime; CAR, carbenicillin; CEP, cephalothin; CFM, cefixime; CFR, cefadroxil; CFX, cefuroxime; CFZ, cefazolin; CHL, chloramphenicol; CIP, ciprofloxacin; COL, colistin/polymyxin E; CPD, cefpodoxime; CPM, cefepime; CTR, ceftriaxone; CTX, cefotaxime; CXT, ceftolozane and tazobactam; CZA, ceftazidime and tazobactam; DOR, doripenem; DOX, doxycycline; ETP, ertapenem; ERY, erythromycin; FOS, fosfomycin; FOX, cefoxitin; FUS, fusidic acid; GEN, gentamicin; IMI, imipenem; LEV, levofloxacin; MEM, meropenem; MIN, minocycline; NAL, nalidixic acid; NIT, nitrofurantoin; NOR, norfloxacin; OFX, ofloxacin; OXA, oxacillin; PIP, piperacillin; POL, polymyxin B; SAM, ampicillin and sulbactam; SXT, sulfamethoxazole and trimethoprim; TCC, ticarcillin and clavulanic acid; TET, tetracycline; TGC, tigecycline; TIC, ticarcillin; TMP, trimethoprim; TOB, tobramycin; TZP, piperacillin and tazobactam

Supplemental Information 2 PRISMA checklist.

Supplemental Information 3 PRISMA Abstract Checklist.

Supplemental Information 4 Rationale & Contribution.

Additional Information and Declarations

Competing Interests

The authors declare that they have no competing interests.

Author Contributions

Mohd Zulkifli Salleh conceived and designed the experiments, performed the experiments, analyzed the data, prepared figures and/or tables, authored or reviewed drafts of the article, and approved the final draft.

Nik Mohd Noor Nik Zuraina performed the experiments, analyzed the data, prepared figures and/or tables, authored or reviewed drafts of the article, and approved the final draft.

Zakuan Zainy Deris analyzed the data, authored or reviewed drafts of the article, and approved the final draft.

Zeehaida Mohamed conceived and designed the experiments, authored or reviewed drafts of the article, and approved the final draft.

Data Availability

The following information was supplied regarding data availability:

All raw data relevant to this systematic review are included in the text, tables, figures, and references.

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
