# Peer review of "Current trends in the epidemiology of multidrug-resistant and beta-lactamase-producing Pseudomonas aeruginosa in Asia and Africa: a systematic review and meta-analysis"

_PeerJ, doi:10.7717/peerj.18986_

## Round 0.1 · original submission · Major Revisions

Dear Authors,

Thank you for submitting your manuscript, titled "Current trends in global epidemiology of multidrug-resistant and beta-lactamase-producing Pseudomonas aeruginosa: a systematic review and meta-analysis," to PeerJ. The reviewers have provided detailed and constructive feedback to improve the manuscript. Based on their reports, we believe the study has the potential to make a valuable contribution but requires significant revisions before it can be considered further. Please send us the point-by-point response to consider further.

Best wishes,
Dr. Nagendran Tharmalingam
Handling Editor

·

Basic reporting

The manuscript is well-organized, and the background information is clearly presented. The literature references are relevant and up to date. However, there are some sections where the language could be refined for better clarity. For example, phrases like "has made treating these infections increasingly challenging" (line 14) and "pooled prevalence rates for multidrug-resistant" (line 22) could be reworded to improve fluency.
The figures and tables are well-labeled and relevant to the content. However, some figure legends, such as those in Table 1, could be more descriptive to provide better context without needing to refer back to the text.

Experimental design

The research question is well-defined, and the systematic review is thorough. The search strategy and inclusion/exclusion criteria are sound and based on standard protocols (PRISMA). The methods section is detailed, making the study replicable. However, the justification for selecting the meta-analysis model (random effects) could be elaborated further, especially in managing significant heterogeneity between studies.

Were there any geographical or population biases in the studies selected? The manuscript would benefit from further discussion of potential limitations in regional representation.

The registration of the study protocol with PROSPERO is a strong aspect of the manuscript, ensuring transparency.

Validity of the findings

The findings are relevant and address the significant issue of multidrug-resistant P. aeruginosa. The global prevalence data provided is insightful. However, the manuscript would benefit from a more detailed discussion on how these findings can guide clinical practice or influence antibiotic stewardship policies.
The statistical analysis is appropriate, but the discussion of heterogeneity is somewhat limited. A more in-depth explanation of how heterogeneity impacts the overall conclusions could strengthen the manuscript.
Some findings, such as the lower resistance rates to last-resort antimicrobials like polymyxin B, could be elaborated further, including hypotheses about the reasons behind these trends.

Additional comments

Could you provide more insight into the limitations regarding data on gender and age distribution across studies? How might these factors influence the overall conclusions, and are there any specific regions where this gap was most pronounced?
There is a need for more focus on the clinical implications of the high variability in resistance rates to cephalosporins and penicillins across regions. How should healthcare systems in different parts of the world adjust their treatment protocols based on your findings?
Some minor English grammar issues were noted, such as the use of "pooled" in certain contexts and occasional awkward phrasing. I suggest a thorough review for language polishing to ensure readability for a broad international audience.

How do the resistance patterns to different antibiotic classes compare across specific hospital settings (ICU vs. non-ICU), and were any significant differences noted in the literature reviewed?
What are the potential implications of the high rates of resistance in specific countries, and how do these align with global efforts to combat antimicrobial resistance?
Considering the significant heterogeneity observed, do you believe a more granular regional meta-analysis would provide more useful insights for local intervention strategies?

·

Basic reporting

The increasing prevalence of antimicrobial-resistant P. aeruginosa presents a severe threat to global health, emphasizing the need for enhanced infection control, effective antibiotic stewardship, and the development of robust, targeted treatment protocols.

Experimental design

This experimental design provides a structured approach for investigating the prevalence and distribution of antimicrobial-resistant P. aeruginosa, enabling robust analysis and relevant conclusions for clinical and public health strategies.

Validity of the findings

If well-conducted, the findings of this review and meta-analysis on antimicrobial-resistant P. aeruginosa would likely be valid, offering critical insights into resistance patterns and global prevalence. However, the generalizability of results depends on geographic diversity, consistency in study definitions, and the methodological rigor of included studies.

Additional comments

Highlight the broader implications of rising P. aeruginosa resistance, especially its impact on healthcare costs, patient outcomes, and infection control practices globally. Emphasizing the threat of MDR and XDR strains to both healthcare systems and public health underscores the urgency of the issue.

·

Basic reporting

Dear Authors,

Your systematic review & Meta-analysis paper entitled "Current trends in global epidemiology of multidrug-resistant and beta-lactamase-producing Pseudomonas aeruginosa: a systematic review and meta-analysis" has been carefully reviewed.

01- This paper is very clear for readers, well designed, and the English language is simple and direct to the point.

02- All the references used in the introduction are relevant to the topic and highlights the main background points related to the topic, including definition of Pseudomonas aeruginosa, its pathogenicity, the disease caused by this germ, and its high resistance to antibiotics, and the MDR and XDR trends of this bacterium. Here I suggest a small point related to the importance of talking about the main virulence factors present in this bacterium, extracellular proteins, metallophores, etc... I suggest to use the following articles as references for this section (Pathogenic Elements):
-- A Review of Pseudomonas aeruginosa Metallophores: Pyoverdine, Pyochelin and Pseudopaline.
-- A novel extracellular protease from Pseudomonas aeruginosa MCM B-327: enzyme production and its partial characterization
-- Extracellular Toxins of Pseudomonas aeruginosa

In addition, In the Introduction section, First Paragraph (Lines 42-48), this paragraph lacks a reference, you are kindly invited to put a reference(s) for this paragraph.

03- The article is well structured, all section are clear for readers, in addition figures and tables are very clear and representative for the main discoveries of the paper. I just have two comments regarding tables legends (in all tables it is Table 1) so you are invited to correct it, and concerning the figures legends it must be below figures.

04- Results presented in this paper are relevant to hypotheses.

Experimental design

01- Since PeerJ is a Multidisciplinary journal and since the paper is well written and structured so the research fits within the Aim and the Scope of the Journal.

02- The research question is well defined, it is also relevant and meaningful. It is stated how research fills and identified knowledge gap.

03- In the present SRMA paper authors made a rigorous literature investigation and analysis using high technical, statistical and ethical standard.

04- Methods used in the present study are described with sufficient detail & information to replicate. But I have one major comment regarding the work here, In the Methodology section, in the paragraph "Search strategy and selection criteria" why you did not use in your search ("Pseudomonas aeruginosa" AND "Drug Resistance") since it will be important for the calculation of the global prevalence of P. aeruginosa.

Validity of the findings

01- The results of this SRMA are of high impact mainly in the medical and the paramedical fields.

02- All underlying data have been provided; they are robust, statistically sound, & controlled.

03- Yes, the conclusion in this SRMA is well stated, and linked to the research question.

Additional comments

I have one additional major comment regarding the 40 studies chosen in this work, it is a little bit strange that there is no evidence data from different continents, mainly from North America, South America, and Australia. In addition there is just one study from Europe (1 study from Italy). This point make the title of the study weak, since in the title you talk about global epidemiology of P. aeruginosa, while in the SRMA 39 studies come from Africa and Asia. This poses a big question regarding your present work.

In addition I find several studies that were not included in your SRMA. for example:

-- Antimicrobial Resistance in Pseudomonas aeruginosa before and during the COVID-19 Pandemic.
-- Epidemiology, Evolution of Antimicrobial Profile and Genomic Fingerprints of Pseudomonas aeruginosa before and during COVID-19: Transition from Resistance to Susceptibility.
-- Clinical Distribution and Drug Resistance of Pseudomonas aeruginosa in Guangzhou, China from 2017 to 2021.
-- Epidemiology and Multidrug Resistance of Pseudomonas aeruginosa and Acinetobacter baumanni Isolated from Clinical Samples in Ethiopia

·

Basic reporting

In the abstract colistin/polymyxin E were included, however no indication about the use of some new antibiotics or their advice or recommendations
the abstract has to include a conclusion

Experimental design

The missing part is the source of the samples as you cannot include environmental samples and compare with clinical ones, or wound samples with pus
In my opinion some important studies were excluded by the selection way

Validity of the findings

1-The estimated global prevalence of P. aeruginosa infections among various age groups (line 170) without any results indicated according to the age. why only 26 studies
2-Figure 3 need to be corrected, as some mistakes appears in the way the values in x axis were written, as the indicated values if representing Y axis values were not important . I would like to know the reason for error bars and if you would like to include them, as this means that you calculated the sum of all sample number under each antibiotic in all 40 studies and then the SD. this point has to be clarified
3- line 254 you cannot write globally based on your data
4-If you would like to include categories of the infection as in line 326, all the results must be categorized according to the source
5-line 330, data representing 6 studied in Pakistan? isolated from the same source of samples? or different sources
6-342 How can you compare ear samples with sewage samples in Nigeria?
7-In line 392, with nearly half of P. aeruginosa strains worldwide now showing resistance to multiple
antibiotics. ( this data in the results of this study or you have to add a reference)
8-line 412 While novel agents like ceftolozane-tazobactam and ceftazidime-avibactam have expanded treatment options, polymyxins often remain the only effective option in certain cases (P. et al. 2019). percentages should be included and also some results and more references
8- In line 437 In our meta-analysis, the global pooled prevalence of ESBL-producing P.
aeruginosa was 33.4% (95% CI: 23.6–43.2), while the prevalence of MBL-producing strains was
found to be 16.0% (95% CI: 9.8–22.3). this sentence has to be discussed for reasons
9-Conclusion has to be rewritten including areas and countries, antibiotics used and recommendations

Additional comments

figure legends missing important data
The discussion has to be more summarized and concentrate on main points and comparisons
I didn;t find any comparison with a previous publication with the same concept by finding resistances in different areas

---

## Round 0.2 · Minor Revisions

Dear Authors,

Thank you for submitting your revised work. Our peers have suggested a minor revision. Please address the issues mentioned in a point-by-point response.

We are looking forward to reading your revised work.

Best wishes,

Dr. Nagendran Tharmalingam
Handling Editor

·

Basic reporting

Dear Authors,

Your revised version of the systematic review & Meta-analysis paper entitled "Current trends in the epidemiology of multidrug-resistant and beta-lactamase-producing Pseudomonas aeruginosa: a systematic review and meta-analysis" has been carefully reviewed.

01- This paper is very clear for readers, well designed, and the English language is simple and direct to the point.

02- All the references used in the introduction are relevant to the topic and highlights the main background points related to the topic, including definition of Pseudomonas aeruginosa, its pathogenicity, the importance of metallophores, the disease caused by this germ, and its high resistance to antibiotics, and the MDR and XDR trends of this bacterium.

03- The article is well structured, all sections are clear for readers, in addition figures and tables are very clear and representative for the main discoveries of the paper.

04- Results presented in this paper are relevant to hypotheses.

Experimental design

01- Since PeerJ is a Multidisciplinary journal and since the paper is well written and structured so the research fits within the Aim and the Scope of the Journal.

02- The research question is well defined, it is also relevant and meaningful. It is stated how research fills and identified knowledge gap.

03- In the present SRMA paper authors made a rigorous literature investigation and analysis using high technical, statistical and ethical standard.

04- Methods used in the present study are described with sufficient detail & information to replicate.

Validity of the findings

01- The results of this SRMA are of high impact mainly in the medical and the paramedical fields.

02- - All underlying data have been provided; they are robust, statistically sound, & controlled.

03- Yes, the conclusion in this SRMA is well stated, and linked to the research question.

Additional comments

I just have a minor comment regarding the title, authors can put that this SRMA is representative for Asia & Africa

·

Basic reporting

In abstract:
Using a random-effects model, our meta-analysis estimated that the overall prevalence of P. aeruginosa in the global population over the past five years was 22.9% (95% CI: 14.4–31.4).

You cannot write globally for a study in restricted area

Experimental design

One sentence concerning environmental samples has to be included to indicated in the exclusion criteria

Validity of the findings

You cannot write globally for a study in restricted area

Additional comments

Most of the comments were answered but still data in relation to the area were the study carried out cannot indicate a global results as other data could be obtained in other areas

---

## Round 0.3 · accepted · Accept

Dear Authors,

Thank you for submitting your revised work. I gladto inform you that your work has been accepted for publish in PeerJ. The production office will contact you for typeset-related queries.

We are looking forward to seeing your future submissions.

Best wishes.
Dr. Nagendran Tharmalingam